POINT OF VIEW

# How open science helps researchers succeed

**Abstract** Open access, open data, open source and other open scholarship practices are growing in popularity and necessity. However, widespread adoption of these practices has not yet been achieved. One reason is that researchers are uncertain about how sharing their work will affect their careers. We review literature demonstrating that open research is associated with increases in citations, media attention, potential collaborators, job opportunities and funding opportunities. These findings are evidence that open research practices bring significant benefits to researchers relative to more traditional closed practices.

ERIN C MCKIERNAN[*], PHILIP E BOURNE, C TITUS BROWN, STUART BUCK, AMYE KENALL, JENNIFER LIN, DAMON MCDOUGALL, BRIAN A NOSEK, KARTHIK RAM, COURTNEY K SODERBERG, JEFFREY R SPIES, KAITLIN THANEY, ANDREW UPDEGROVE, KARA H WOO AND TAL YARKONI

**\*For correspondence:**
emckiernan@ciencias.unam.mx

## Introduction

Recognition and adoption of open research practices is growing, including new policies that increase public access to the academic literature (open access; *Björk et al., 2014*; *Swan et al., 2015*) and encourage sharing of data (open data; *Heimstädt et al., 2014*; *Michener, 2015*; *Stodden et al., 2013*), and code (open source; *Stodden et al., 2013*; *Shamir et al., 2013*). Such policies are often motivated by ethical, moral or utilitarian arguments (*Suber, 2012*; *Willinsky, 2006*), such as the right of taxpayers to access literature arising from publicly-funded research (*Suber, 2003*), or the importance of public software and data deposition for reproducibility (*Poline et al., 2012*; *Stodden, 2011*; *Ince et al., 2012*). Meritorious as such arguments may be, however, they do not address the practical barriers involved in changing researchers' behavior, such as the common perception that open practices could present a risk to career advancement. In the present article, we address such concerns and suggest that the benefits of open practices outweigh the potential costs.

We take a researcher-centric approach in outlining the benefits of open research practices. Researchers can use open practices to their advantage to gain more citations, media attention, potential collaborators, job opportunities and funding opportunities. We address common myths about open research, such as concerns about the rigor of peer review at open access journals, risks to funding and career advancement, and forfeiture of author rights. We recognize the current pressures on researchers, and offer advice on how to practice open science within the existing framework of academic evaluations and incentives. We discuss these issues with regard to four areas – publishing, funding, resource management and sharing, and career advancement – and conclude with a discussion of open questions.

## Publishing

### *Open publications get more citations*

There is evidence that publishing openly is associated with higher citation rates (*Hitchcock, 2016*). For example, Eysenbach reported that articles published in the *Proceedings of the National Academy of Sciences* (*PNAS*) under their open access (OA) option were twice as likely to be cited within 4–10 months and nearly three times as likely to be cited 10–16 months after publication than non-OA articles published

**Figure 1.** Open access articles get more citations. The relative citation rate (OA: non-OA) in 19 fields of research. This rate is defined as the mean citation rate of OA articles divided by the mean citation rate of non-OA articles. Multiple points for the same discipline indicate different estimates from the same study, or estimates from several studies. References by discipline: Agricultural studies (*Kousha and Abdoli, 2010*); Physics/astronomy (*Gentil-Beccot et al., 2010*; *Harnad and Brody, 2004*; *Metcalfe, 2006*); Medicine (*Sahu et al., 2005*; *Xu et al., 2011*); Computer science (*Lawrence, 2001*); Sociology/social sciences (*Hajjem et al., 2006*; *Norris et al., 2008*; *Xu et al., 2011*); Psychology (*Hajjem et al., 2006*); Political science (*Hajjem et al., 2006*; *Antelman, 2004*; *Atchison and Bull, 2015*); Management (*Hajjem et al., 2006*); Law (*Donovan et al., 2015*; *Hajjem et al., 2006*); Economics (*Hajjem et al., 2006*; *McCabe and Snyder, 2015*; *Norris et al., 2008*; *Wohlrabe, 2014*); Mathematics (*Antelman, 2004*; *Davis and Fromerth, 2007*; *Norris et al., 2008*); Health (*Hajjem et al., 2006*); Engineering (*Antelman, 2004*; *Koler-Povh et al., 2014*); Philosophy (*Antelman, 2004*); Education (*Hajjem et al., 2006*; *Zawacki-Richter et al., 2010*); Business (*Hajjem et al., 2006*; *McCabe and Snyder, 2015*); Communication studies (*Zhang, 2006*); Ecology (*McCabe and Snyder, 2014*; *Norris et al., 2008*); Biology (*Frandsen, 2009b*; *Hajjem et al., 2006*; *McCabe and Snyder, 2014*).

in the same journal (*Eysenbach, 2006*). Hajjem and colleagues studied over 1.3 million articles published in 10 different disciplines over a 12-year period and found that OA articles had a 36–172% advantage in citations over non-OA articles (*Hajjem et al., 2006*). While some controlled studies have failed to find a difference in citations between OA and non-OA articles or attribute differences to factors other than access (*Davis, 2011*; *Davis et al., 2008*; *Frandsen, 2009a*; *Gaulé and Maystre, 2011*; *Lansingh and Carter, 2009*), a larger number of studies confirm the OA citation advantage. Of 70 studies registered as of June 2016 in the Scholarly Publishing and Academic Resources Coalition (SPARC) Europe database of citation

studies, 46 (66%) found an OA citation advantage, 17 (24%) found no advantage, and 7 (10%) were inconclusive (*SPARC Europe, 2016*). Numerical estimates of the citation advantage in two reviews range from -5% to 600% (*Swan, 2010*) and 25% to 250% (*Wagner, 2010*). The size of the advantage observed is often dependent on discipline (*Figure 1*). Importantly, the OA citation advantage can be conferred regardless of whether articles are published in fully OA journals, subscription journals with OA options (hybrid journals), or self-archived in open repositories (*Eysenbach, 2006*; *Hajjem et al., 2006*; *Gargouri et al., 2010*; *Research Information Network, 2014*; *Wang et al., 2015*; *Swan, 2010*; *Wagner, 2010*).

Moreover, at least in some cases, the advantage is not explained by selection bias (i.e., authors deliberately posting their better work to open platforms), as openly archived articles receive a citation advantage regardless of whether archiving is initiated by the author or mandated by an institution or funder (*Gargouri et al., 2010*; *Xia and Nakanishi, 2012*).

### Open publications get more media coverage

One way for researchers to gain visibility is for their publications to be shared on social media and covered by mainstream media outlets. There is evidence that publishing articles openly can help researchers get noticed. A study of over 2,000 articles published in *Nature Communications* showed that those published openly received nearly double the number of unique tweeters and Mendeley readers as closed-access articles (*Adie, 2014a*). A similar study of over 1,700 *Nature Communications* articles found that OA articles receive 2.5–4.4 times the number of page views, and garnered more social media attention via Twitter and Facebook than non-OA articles (*Wang et al., 2015*). There is tentative evidence that news coverage confers a citation advantage. For example, a small quasi-experimental 1991 study found that articles covered by the *New York Times* received up to 73% more citations that those not covered (*Phillips et al., 1991*). A 2003 correlational study supported these results, reporting higher citation rates for articles covered by the media (*Kiernan, 2003*).

### Prestige and journal impact factor

As Sydney Brenner wrote in 1995, ''...*what matters absolutely is the scientific content of a paper and...nothing will substitute for either knowing it or reading it*'' (*Brenner, 1995*). Unfortunately, academic institutions often rely on proxy metrics, like journal impact factor (IF), to quickly evaluate researchers' work. The IF is a flawed metric that correlates poorly with the scientific quality of individual articles (*Brembs et al., 2013*; *Neuberger and Counsell, 2002*; *PLOS Medicine Editors, 2006*; *Seglen, 1997*). In fact, several of the present authors have signed the San Francisco Declaration on Research Assessment (SF-DORA) recommending IF not be used as a research evaluation metric (*American Society for Cell Biology, 2013*). However, until institutions cease using IF in evaluations, researchers will understandably be concerned about the IF of journals in which they publish. In author surveys, researchers repeatedly rank IF and associated journal reputation as among the most important factors they consider when deciding where to publish (*Nature Publishing Group, 2015*; *Solomon, 2014*). Researchers are also aware of the associated prestige that can accompany publication in high-IF journals such as *Nature* or *Science*. Thus, OA advocates should recognize and respect the pressures on researchers to select publishing outlets based, at least in part, on IF.

Fortunately, concerns about IF need not prevent researchers from publishing openly. For one thing, the IFs of indexed OA journals are steadily approaching those of subscription journals (*Björk and Solomon, 2012*). In the 2012 Journal Citation Report, over 1,000 (13%) of the journals listed with IFs were OA (*Gunasekaran and Arunachalam, 2014*). Of these OA journals, thirty-nine had IFs over 5.0 and nine had IFs over 10.0. Examples of OA journals in the biological and medical sciences with moderate to high 2015 IFs include *PLOS Medicine* (13.6), *Nature Communications* (11.3), and BioMed Central's *Genome Biology* (11.3). The Cofactor Journal Selector Tool allows authors to search for OA journals with an IF (*Cofactor Ltd, 2016*). We reiterate that our goal in providing such information is not to support IF as a valid measure of scholarly impact, but to demonstrate that researchers do not have to choose between IF and OA when making publishing decisions.

In addition, many subscription-based high-IF journals offer authors the option to pay to make their articles openly accessible. While one can debate the long-term viability and merits of a model that allows publishers to effectively reap both reader-paid and author-paid charges (*Björk, 2012*), in the short term, researchers who wish to publish their articles openly in traditional journals can do so. Researchers can also publish in high-IF subscription journals and self-archive openly (see section "Publish where you want and archive openly"). We hope that in the next few years, use of IF as a metric will diminish or cease entirely, but in the meantime, researchers have options to publish openly while still meeting any IF-related evaluation and career advancement criteria.

### Rigorous and transparent peer review

Unlike most subscription journals, several OA journals have open and transparent peer review processes. Journals such as *PeerJ* and Royal Society's *Open Science* offer reviewers the

opportunity to sign their reviews and offer authors the option to publish the full peer review history alongside their articles. In 2014, *PeerJ* reported that ∼40% of reviewers sign their reports and ∼80% of authors choose to make their review history public (*PeerJ Staff, 2014*). BioMed Central's *GigaScience*, all the journals in BMC's medical series, Copernicus journals, *F1000Research*, and MDPI's *Life* require that reviewer reports be published, either as part of a prepublication review process, or subsequent to publication. Some studies suggest open peer review may produce reviews of higher quality, including better substantiated claims and more constructive criticisms, compared to closed review (*Kowalczuk et al., 2013*; *Walsh et al., 2000*). Additional studies have also argued that transparent peer review processes are linked to measures of quality (*Wicherts, 2016*). Other studies have reported no differences in the quality of open versus closed reviews (*van Rooyen et al., 1999*; *van Rooyen et al., 2010*). More research in this area is needed.

Unfortunately, the myth that OA journals have poor or non-existent peer review persists. This leads many to believe that OA journals are low quality and causes researchers to be concerned that publishing in these venues will be considered less prestigious in academic evaluations. To our knowledge, there has been no controlled study comparing peer review in OA versus subscription journals. Studies used by some to argue the weakness of peer review at OA journals, such as the John Bohannon 'sting' (*Bohannon, 2013*) in which a fake paper was accepted by several OA journals, have been widely criticized in the academic community for poor methodology, including not submitting to subscription journals for comparison (*Joseph, 2013*; *Redhead, 2013*). In fact, Bohannon admitted, *''Some open-access journals that have been criticized for poor quality control provided the most rigorous peer review of all''*. He cites *PLOS ONE* as an example, saying it was the only journal to raise ethical concerns with his submitted work (*Bohannon, 2013*).

Subscription journals have not been immune to problems with peer review. In 2014, Springer and IEEE retracted over 100 published fake articles from several subscription journals (*Van Noorden, 2014*; *Springer, 2014*). Poor editorial practices at one SAGE journal opened the door to peer review fraud that eventually led 60 articles to be retracted (*Bohannon, 2014*; *Journal of Vibration and Control, 2014*). Similar issues in other subscription journals have been documented by Retraction Watch (*Oransky and Marcus, 2016*). Problems with peer review thus clearly exist, but are not exclusive to OA journals. Indeed, large-scale empirical analyses indicate that the reliability of the traditional peer review process itself leaves much to be desired. Bornmann and colleagues reviewed 48 studies of inter-reviewer agreement and found that the average level of agreement was low (mean ICC of .34 and Cohen's kappa of .17) – well below what what would be considered adequate in psychometrics or other fields focused on quantitative assessment (*Bornmann et al., 2010*). Opening up peer review, including allowing for real-time discussions between authors and reviewers, could help address some of these issues.

Over time, we expect that transparency will help dispel the myth of poor peer review at OA journals, as researchers read reviews and confirm that the process is typically as rigorous as that of subscription journals. Authors can use open reviews to demonstrate to academic committees the rigorousness of the peer review process in venues where they publish, and highlight reviewer comments on the importance of their work. Researchers in their capacity as reviewers can also benefit from an open approach, as this allows them to get credit for this valuable service. Platforms like Publons let researchers create reviewer profiles to showcase their work (*Publons, 2016*).

## Publish where you want and archive openly

Some researchers may not see publishing in OA journals as a viable option, and may wish instead to publish in specific subscription journals seen as prestigious in their field. Importantly, there are ways to openly share work while still publishing in subscription journals.

**Preprints**: Authors may provide open access to their papers by posting them as preprints prior to formal peer review and journal publication. Preprints servers are both free for authors to post and free for readers. Several archival preprint servers exist covering different subject areas (*Table 1*). (Note: The list in *Table 1* is not all-inclusive; there are many other servers and institutional repositories that also accept preprints).

Many journals allow posting of preprints, including *Science*, *Nature*, and *PNAS*, as well as most OA journals. Journal preprint policies can be checked via Wikipedia (*Wikipedia, 2016*) and SHERPA/RoMEO (*SHERPA/RoMEO, 2016*). Of

**Table 1.** Preprint servers and general repositories accepting preprints.

| Preprint server or repository* | Subject areas | Repository open source? | Public API? | Can leave feedback?[†] | Third party persistent ID? |
|---|---|---|---|---|---|
| arXiv arxiv.org | physics, mathematics, computer science, quantitative biology, quantitative finance, statistics | No | Yes | No | No[‡] |
| bioRxiv biorxiv.org | biology, life sciences | No | No | Yes | Yes (DOI) |
| CERN document server cds.cern.ch | high-energy physics | Yes (GPL) | Yes | No | No |
| Cogprints cogprints.org | psychology, neuroscience, linguistics, computer science, philosophy, biology | No | Yes | No | No |
| EconStor econstor.eu | economics | No | Yes | No | Yes (Handle) |
| e-LiS eprints.rclis.org | library and information sciences | No[§] | Yes | No | Yes (Handle) |
| figshare figshare.com | general repository for all disciplines | No | Yes | Yes | Yes (DOI) |
| Munich Personal RePEc Archive mpra.ub.uni-muenchen.de | economics | No[¶] | Yes | No | No |
| Open Science Framework osf.io | general repository for all disciplines | Yes (Apache 2) | Yes | Yes | Yes (DOI/ARK) |
| PeerJ Preprints peerj.com/archives-preprints | biological, life, medical, and computer sciences | No | Yes | Yes | Yes (DOI) |
| PhilSci Archive philsci-archive.pitt.edu | philosophy of science | No** | Yes | No | No |
| Self-Journal of Science www.sjscience.org | general repository for all disciplines | No | No | Yes | No |
| Social Science Research Network ssrn.com | social sciences and humanities | No | No | Yes | Yes (DOI) |
| The Winnower thewinnower.com | general repository for all disciplines | No | No | Yes | Yes (DOI)[††] |
| Zenodo zenodo.org | general repository for all disciplines | Yes (GPLv2) | Yes | No | Yes (DOI) |

* All these servers and repositories are indexed by Google Scholar.

[†] Most, if not all, of those marked 'Yes' require some type of login or registration to leave comments.

[‡] arXiv provides internally managed persistent identifiers.

[§] e-LiS is built on open source software (EPrints), but the repository itself, including modifications to the code, plugins, etc. is not open source.

[¶] MPRA is built on open source software (EPrints), but the repository itself, including modifications to the code, plugins, etc. is not open source.

** PhilSci Archive is built on open source software (EPrints), but the repository itself, including modifications to the code, plugins, etc. is not open source.

[††] The Winnower charges a $25 fee to assign a DOI.

the over 2,000 publishers in the SHERPA/RoMEO database, 46% explicitly allow preprint posting. Preprints can be indexed in Google Scholar and cited in the literature, allowing authors to accrue citations while the paper is still in review. In one extreme case, one of the present authors (CTB) published a preprint that has received over 50 citations in three years (*Brown et al., 2012*), and was acknowledged in NIH grant reviews.

In some fields, preprints can establish scientific priority. In physics, astronomy, and

mathematics, preprints have become an integral part of the research and publication workflow (**Brown, 2001**; **Larivière et al., 2014**; **Gentil-Beccot et al., 2010**). Physics articles posted as preprints prior to formal publication tend to receive more citations than those published only in traditional journals (**Gentil-Beccot et al., 2010**; **Schwarz and Kennicutt Jr, 2004**; **Metcalfe, 2006**). Unfortunately, because of the slow adoption of preprints in the biological and medical sciences, few if any studies have been conducted to examine citation advantage conferred by preprints in these fields. However, the growing number of submissions to the quantitative biology section of arXiv, as well as to dedicated biology preprint servers such as bioRxiv and PeerJ PrePrints, should make such studies feasible. Researchers have argued for increased use of preprints in biology (**Desjardins-Proulx et al., 2013**). The recent Accelerating Science and Publication in biology (ASAPbio) meeting demonstrates growing interest and support for life science preprints from researchers, funders, and publishers (**Berg et al., 2016**; **ASAPbio, 2016**).

**Postprints:** Authors can also archive articles on open platforms after publication in traditional journals (postprints). SHERPA/RoMEO allows authors to check policies from over 2,200 publishers, 72% of which allow authors to archive postprints, either in the form of the authors' accepted manuscript post-peer review, or the publisher's formatted version, depending on the policy (**SHERPA/RoMEO, 2016**). Of notable example is *Science*, which allows authors to immediately post the accepted version of their manuscript on their website, and post to larger repositories like PubMed Central six months after publication. The journal *Nature* likewise allows archiving of the accepted article in open repositories six months after publication.

If the journal in which authors publish does not formally support self-archiving, authors can submit an author addendum that allows them to retain rights to post a copy of their article in an open repository. The Scholarly Publishing and Academic Resources Coalition (SPARC) provides a template addendum, as well as information on author rights (**SPARC, 2016**). The Scholar's Copyright Addendum Engine helps authors generate a customized addendum to send to publishers (**Science Commons, 2016**). Not all publishers will accept author addenda, but some are willing to negotiate the terms of their publishing agreements.

## Retain author rights and control reuse with open licenses

To make their findings known to the world, scientists have historically forfeited ownership of the products of their intellectual labor by signing over their copyrights or granting exclusive reuse rights to publishers. In contrast, authors publishing in OA journals retain nearly all rights to their manuscripts and materials. OA articles are typically published under Creative Commons (CC) licenses, which function within the legal framework of copyright law (**Creative Commons, 2016**). Under these licenses, authors retain copyright, and simply grant specific (non-exclusive) reuse rights to publishers, as well as other users. Moreover, CC licenses require attribution, which allows authors to receive credit for their work and accumulate citations. Licensors can specify that attribution include not just the name of the author(s) but also a link back to the original work. Authors submitting work to an OA journal should review its submission rules to learn what license(s) the journal permits authors to select.

If terms of a CC license are violated by a user, the licensor can revoke the license and, if the revocation is not honored, take legal action to enforce their copyright. There are several legal precedents upholding CC licenses, including: (1) Adam Curry v. Audax Publishing (**Court of Amsterdam, 2006**; **Garlick, 2006a**); (2) Sociedad General de Autores y Editores (SGAE) v. Ricardo Andrés Utrera Fernández (**Juzgado de Primera Instancia Número Seis de Badajoz, España, 2006**; **Garlick, 2006b**); and (3) Gerlach v. Deutsche Volksunion (DVU) (**Linksvayer, 2011**). Through open licensing, researchers thus retain control over how their work is read, shared, and used by others.

An emerging and interesting development is the adoption of rights-retention open access policies (**Harvard Open Access Project, 2016**). To date, such policies have been adopted by at least 60 schools and institutions worldwide, including some in Canada, Iceland, Kenya, Saudi Arabia, and U.S. universities like Harvard (**Harvard Library, Office for Scholary Communication, 2016**) and MIT (**MIT Libraries, Scholarly Publishing, 2016**). These policies involve an agreement by the faculty to grant universities non-exclusive reuse rights on future published works. By putting such a policy in place prior to publication, faculty work can be openly archived without the need to negotiate with publishers to retain or recover rights; open is the default. We

expect to see adoption of such policies grow in coming years.

### Publish for low-cost or no-cost

Researchers often cite high costs, primarily in the form of article processing charges (APCs), as a barrier to publishing in OA journals. While some publishers – subscription as well as OA – do charge steep fees (*Lawson, 2016*; *Wellcome Trust, 2016c*), many others charge nothing at all. In a 2014 study of 1,357 OA journals, 71% did not request any APC (*West et al., 2014*). A study of over 10,300 OA journals from 2011 to 2015 likewise found 71% did not charge (*Crawford, 2016*). Eigenfactor.org maintains a list of hundreds of no-fee OA journals across fields (*Eigenfactor Project, 2016*). Researchers can also search for no-cost OA journals using the Cofactor Journal Selector tool (*Cofactor Ltd, 2016*). Notable examples of OA journals which do not currently charge authors to publish include *eLife*, Royal Society's *Open Science*, and all journals published by consortiums like Open Library of Humanities and SCOAP[3]. The Scientific Electronic Library Online (SciELO) and the Network of Scientific Journals in Latin America, the Caribbean, Spain, and Portugal (Redalyc), each host over 1,000 journals that provide free publishing for authors.

Many other OA journals charge minimal fees, with the average APC around $665 USD (*Crawford, 2016*). At *PeerJ*, for example, a one-time membership fee of $199 USD allows an author to publish one article per year for life, subject to peer review. (Note: Since *PeerJ* requires the membership fee to be paid for each author up to 12 authors, the maximum cost of an article would be $2,388 USD. However, this is a one-time fee, after which subsequent articles for the same authors would be free.) Most Pensoft OA journals charge around €100–400 (~$115–460 USD), while a select few are free. Ubiquity Press OA journals charge an average APC of £300 (~$500 USD), with their open data and software metajournals charging £100 (~$140 USD). Cogent's OA journals all function on a flexible payment model, with authors paying only what they are able based on their financial resources. Importantly, most OA journals do not charge any additional fees for submission or color figures. These charges, as levied by many subscription publishers, can easily sum to hundreds or thousands of dollars (e.g. in Elsevier's *Neuron* the first color figure is $1,000 USD, while each additional one is $275). Thus, publishing in OA journals need not be any more expensive than publishing in traditional journals, and in some cases, may cost less.

The majority of OA publishers charging higher publication fees (e.g., PLOS or Frontiers, which typically charge upwards of $1,000 USD per manuscript) offer fee waivers upon request for authors with financial constraints. Policies vary by publisher, but frequently include automatic full waivers for authors from low-income countries, and partial waivers for those in lower-middle-income countries. Researchers in any country can request a partial or full waiver if they cannot pay. Some publishers, such as BioMed Central, F1000, Hindawi, and PeerJ, have membership programs through which institutions pay part or all of the APC for affiliated authors. Some institutions also have discretionary funds for OA publication fees. Increasingly, funders are providing OA publishing funds, or allowing researchers to write these funds into their grants. PLOS maintains a searchable list of both institutions and funders that support OA publication costs (*Public Library of Science, 2016*). Finally, as discussed previously in the section "Publish where you want and archive openly", researchers can make their work openly available for free by self-archiving preprints or postprints.

## Funding

### Awards and special funding

For academics in many fields, securing funding is essential to career development and success of their research program. In the last three years, new fellowships and awards for open research have been created by multiple organizations (*Table 2*). While there is no guarantee that these particular funding mechanisms will be maintained, they are a reflection of the changing norms in science, and illustrate the increasing opportunities to gain recognition and resources by sharing one's work openly.

### Funder mandates on article and data sharing

Increasingly, funders are not only preferring but mandating open sharing of research. The United States National Institutes of Health (NIH) has been a leader in this respect. In 2008, the NIH implemented a public access policy, requiring that all articles arising from NIH-funded projects be deposited in the National Library of Medicine's open repository, PubMed Central, within one year of publication (*Rockey, 2012*). NIH

**Table 2.** Special funding opportunities for open research, training, and advocacy.

| Funding | Description | URL |
|---|---|---|
| Shuttleworth Foundation Fellowship Program | funding for researchers working openly on diverse problems | shuttleworthfoundation.org/fellows/ |
| Mozilla Fellowship for Science | funding for researchers interested in open data and open source | www.mozillascience.org/fellows |
| Leamer-Rosenthal Prizes for Open Social Science (UC Berkeley and John Templeton Foundation) | rewards social scientists for open research and education practices | www.bitss.org/prizes/leamer-rosenthal-prizes/ |
| OpenCon Travel Scholarship (Right to Research Coalition and SPARC) | funding for students and early-career researchers to attend OpenCon, and receive training in open practices and advocacy | www.opencon2016.org/ |
| Preregistration Challenge (Center for Open Science) | prizes for researchers who publish the results of a preregistered study | cos.io/prereg/ |
| Open Science Prize (Wellcome Trust, NIH, and HHMI) | funding to develop services, tools, and platforms that will increase openness in biomedical research | www.openscienceprize.org/ |

also requires that projects receiving $500K or more per year in direct costs include a data management plan that specifies how researchers will share their data (*National Institutes of Health, 2003*). NIH intends to extend its data sharing policy to a broader segment of its portfolio in the near future. Since 2011, the United States National Science Foundation (NSF) has also encouraged sharing data, software, and other research outputs (*National Science Foundation, 2011*). All NSF investigators are required to submit a plan, specifying data management and availability. In 2015, U.S. government agencies, including the NSF, Centers for Disease Control and Prevention (CDC), Department of Defense (DoD), National Aeronautics and Space Administration (NASA), and more announced plans to implement article and data sharing requirements in response to the White House Office of Science and Technology (OTSP) memo on public access (*Holdren, 2013*). A crowd-sourced effort has collected information on these agency policies and continues to be updated (*Whitmire et al., 2015*).

Several governmental agencies and charitable foundations around the world have implemented even stronger open access mandates. For example, the Wellcome Trust's policy states that articles from funded projects must be made openly available within six months of publication, and where it provides publishing fee support, specifically requires publication under a Creative Commons Attribution (CC BY) license (*Wellcome Trust, 2016b*). The Netherlands Organization for Scientific Research (NWO) requires that all manuscripts reporting results produced using public funds must be made immediately available (*NWO, 2016*). Similar policies are in place at CERN (*CERN, 2014*), the United Nations Educational, Scientific and Cultural Organization (*UNESCO, 2013*), and the Bill & Melinda Gates Foundation (*Bill & Melinda Gates Foundation, 2015*) among others, and are increasingly covering data sharing. Funders recognize that certain types of data, such as clinical records, are sensitive and require special safeguards to permit sharing while protecting patient privacy. The Expert Advisory Group on Data Access (EAGDA) was recently established as a collaboration between the Wellcome Trust, Cancer Research UK, the Economic and Social Research Council, and the Medical Research Council to advise funders on best practices for creating data sharing policies for human research (*Wellcome Trust, 2016a*).

Researchers can check article and data sharing policies of funders in their country via SHERPA/JULIET (*SHERPA/JULIET, 2016*). Bio-Sharing also maintains a searchable database of data management and sharing policies from both funders and publishers worldwide (*Biosharing.org, 2016*). Internationally, the number of open access policies has been steadily increasing over the last decade (*Figure 2*). Some funders, including the NIH and Wellcome Trust, have begun suspending or withholding funds if researchers do not meet their policy requirements (*National Institutes of Health, 2012*; *Van Noorden, 2014*; *Wellcome Trust, 2012*). Thus, researchers funded by a wide variety of sources will soon be not just encouraged but required to engage in open practices to receive and retain funding. Those already engaging in these practices will likely have a competitive advantage.

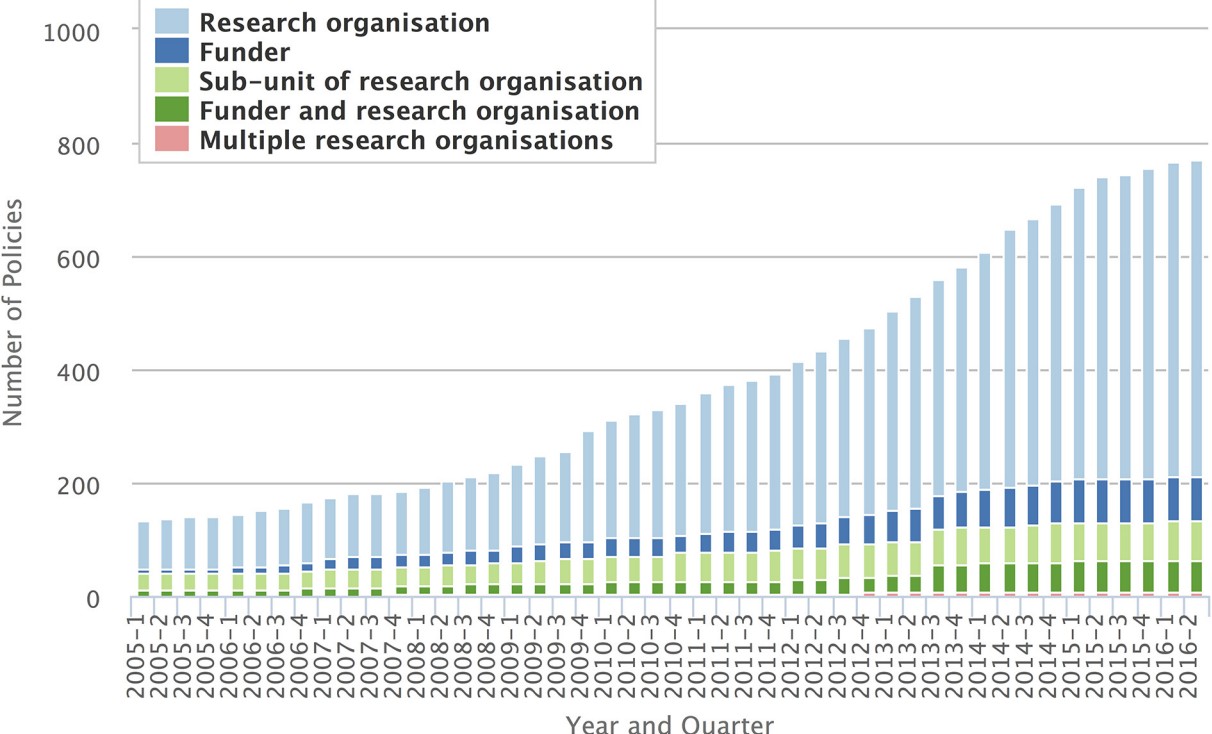

**Figure 2.** Increase in open access policies. The number of open access policies registered in ROARMAP (roarmap.eprints.org) has increased over the last decade. Data are broken down by type of organization: research organization (e.g., a university or research institution); funder; subunit of research organization (e.g. a library within a university); funder and research organization; multiple research organizations (e.g., an organization with multiple research centers, such as Max Planck Society). Figure used with permission from Stevan Harnad.

## Resource management and sharing

In our researcher-centric approach, the rationale for data sharing based on funder mandates could be understood simply as 'funders want you to share, so it is in your interest to do so'. That may be a compelling but dissatisfying reason to practice openly. Fortunately, there are other compelling reasons to share.

*Documentation and reproducibility benefits*

First, submitting data and research materials to an independent repository ensures preservation and accessibility of that content in the future - both for one's own access and for others. This is a particular benefit for responding to requests for data or materials by others. Preparation of research materials for sharing during the active phase of the project is much easier than reconstructing work from years earlier. Second, researchers who plan to release their data, software, and materials are likely to engage in behaviors that are easy to skip in the short-term but have substantial benefits in the long-term,

such as clear documentation of the key products of the research. Besides direct benefits for one-self in facilitating later reuse, such practices increase the reproducibility of published findings and the ease with which other researchers can use, extend, and cite that work (*Gorgolewski and Poldrack, 2016*). Finally, sharing data and materials signals that researchers value transparency and have confidence in their own research.

*Gain more citations and visibility by sharing data*

Data sharing also confers a citation advantage. *Piwowar and Vision (2013)* analyzed over 10,000 studies with gene expression microarray data published in 2001–2009, and found an overall 9% citation advantage for papers with shared data and advantages around 30% for older studies. *Henneken and Accomazzi (2011)* found a 20% citation advantage for astronomy articles that linked to open datasets. *Dorch et al., 2015* found a 28–50% citation advantage for astrophysics articles, while *Sears (2011)* reported a 35% advantage for

paleoceanography articles with publicly available data. Similar positive effects of data sharing have been described in the social sciences. *Gleditsch et al., 2003* found that articles in the *Journal of Peace Research* offering data in any form – either through appendices, URLs, or contact addresses – were cited twice as frequently on average as articles with no data but otherwise equivalent author credentials and article variables. Studies with openly published code are also more likely to be cited than those that do not open their code (*Vandewalle, 2012*). In addition to more citations, *Pienta et al., 2010* found that data sharing is associated with higher publication productivity. Across over 7,000 NSF and NIH awards, they reported that research projects with archived data produced a median of 10 publications, versus only 5 for projects without archived data.

Importantly, citation studies may underestimate the scientific contribution and resulting visibility associated with resource sharing, as many data sets and software packages are published as stand-alone outputs that are not associated with a paper but may be widely reused. Fortunately, new outlets for data and software papers allow researchers to describe new resources of interest without necessarily reporting novel findings (*Chavan and Penev, 2011*; *Gorgolewski et al., 2013*). There is also a growing awareness that data and software are independent, first class scholarly outputs, that need to be incorporated into the networked research ecosystem. Many open data and software repositories have mechanisms for assigning digital object identifiers (DOIs) to these products. The use of persistent, unique identifiers like DOIs has been recommended by the Joint Declaration of Data Citation Principles to facilitate data citation (*Data Citation Synthesis Group, 2014*). Researchers can register for a unique Open Researcher and Contributor ID (ORCID) (*Haak et al., 2012*) to track their research outputs, including datasets and software, and build a richer profile of their contributions. Together, these developments should support efforts to ''make data count'', further incentivize sharing, and ensure that data generators and software creators receive greater credit for their work (*Kratz and Strasser, 2015*).

In summary, data and software sharing benefits researchers both because it is consistent with emerging mandates, and because it signals credibility and engenders good research practices that can reduce errors and promote reuse, extension, and citation.

## Career advancement

### Find new projects and collaborators

Research collaborations are essential to advancing knowledge, but identifying and connecting with appropriate collaborators is not trivial. Open practices can make it easier for researchers to connect with one another by increasing the discoverability and visibility of one's work, facilitating rapid access to novel data and software resources, and creating new opportunities to interact with and contribute to ongoing communal projects. For example, in 2011, one of the present authors (BAN) initiated a project to replicate a sample of studies to estimate the reproducibility of psychological science (*Open Science Collaboration, 2012*; *Open Science Collaboration, 2014*). Completing a meaningful number of replications in a single laboratory would have been difficult. Instead, the project idea was posted to a listserv as an open collaboration. Ultimately, more than 350 people contributed, with 270 earning co-authorship on the publication (*Open Science Collaboration, 2015*). Open collaboration enabled distribution of work and expertise among many researchers, and was essential for the project's success. Other projects have used similar approaches to successfully carry out large-scale collaborative research (*Klein et al., 2014*).

Similar principles are the core of the thriving open -source scientific software ecosystem. In many scientific fields, widely used state-of-the-art data processing and analysis packages are hosted and developed openly, allowing virtually anyone to contribute. Perhaps the paradigmatic example is the *scikit-learn* Python package for machine learning (*Pedregosa et al., 2011*), which, in the space of just over five years, has attracted over 500 unique contributors, 20,000 individual code contributions, and 2,500 article citations. Producing a comparable package using a traditional closed-source approach would likely not be feasible, and would, at the very least, have required a budget of tens of millions of dollars. While scikit-learn is clearly an outlier, hundreds of other open-source scientific packages that support much more domain-specific needs depend in a similar fashion on unsolicited community contributions e.g., the NIPY group of projects in neuro-imaging (*Gorgolewski et al., 2016*). Importantly, such contributions not only result in new functionality from which the broader scientific community can benefit, but also regularly provide their respective authors with greater community recognition, and lead to new project and employment opportunities.

## Institutional support of open research practices

Institutions are increasingly recognizing the limitations of journal-level metrics and exploring the potential benefits of article-level and alternative metrics in evaluating the contributions of specific research outputs. In 2013, the American Society for Cell Biology, along with a group of diverse stakeholders in academia, released the San Francisco Declaration on Research Assessment (SF-DORA) (*American Society for Cell Biology, 2013*). The declaration recommends that institutions cease using all journal-level metrics, including journal impact factor (IF), to evaluate research for promotion and tenure decisions, and focus instead on research content. Additional recommendations include recognizing data and software as valuable research products. As of March 2016, over 12,000 individuals and more than 600 organizations have signed SF-DORA in support of the recommendations, including universities from all over the world. The 2015 Higher Education Funding Council for England (HEFCE) report for The Research Excellence Framework (REF) – UK's system for assessing research quality in higher education institutions – also rejects the use of IF and other journal metrics to evaluate researchers for hiring and promotion, and recommends institutions explore a variety of quantitative and qualitative indicators of research impact and ways to recognize sharing of diverse research outputs (*Wilsdon et al., 2015*).

Several U.S. institutions have passed resolutions explicitly recognizing open practices in promotion and tenure evaluations, including Virginia Commonwealth University (*Virginia Commonwealth University Faculty Senate, 2010*) and Indiana University-Purdue University Indianapolis (*Indiana University-Purdue University Indianapolis, 2016*). In 2014, Harvard's School of Engineering and Applied Sciences launched a pilot program to encourage faculty to archive their articles in the university's open repository as part of the promotion and tenure process (*Harvard Library, Office for Scholarly Communication, 2014*). The University of Liège has gone a step further and requires publications to be included in the university's open access repository to be considered for promotion (*University of Liège, 2016*). Explicit statements of the importance of open practices are even starting to appear in faculty job advertisements, such as one from LMU München asking prospective candidates to describe their open research activities (*Schönbrodt, 2016*).

## Discussion

### Open questions

The emerging field of metascience provides some evidence about the value of open practices, but it is far from complete. There are many initiatives aimed at increasing open practices, and not yet enough published evidence about their effectiveness. For example, journals can offer badges to acknowledge open practices such as open data, open materials, and preregistration (*Open Research Badges, 2016*). Initial evidence from a single adopting journal, *Psychological Science*, and a sample of comparison journals suggests that this simple incentive increases data sharing rates from less than 3% to more than 38% (*Kidwell et al., 2016*). More research is needed across disciplines to follow up on this encouraging evidence. UCLA's Knowledge Infrastructures project is an ongoing study that, among other objectives, is learning about data sharing practices and factors that discourage or promote sharing across four collaborative scientific projects (*Borgman et al., 2015*; *Darch et al., 2015*).

Open research advocates often cite reproducibility as one of the benefits of data and code sharing (*Gorgolewski and Poldrack, 2016*). There is a logical argument that having access to the data, code, and materials makes it easier to reproduce the evidence that was derived from that research content. Data sharing correlates with fewer reporting errors, compared to papers with unavailable data (*Wicherts, 2016*), and could be due to diligent data management practices. However, there is not yet direct evidence that open practices per se are a net benefit to research progress. As a first step, the University of California at Riverside and the Center for Open Science have initiated an NSF-supported randomized trial to evaluate the impact of receiving training to use the Open Science Framework for managing, archiving, and sharing lab research materials and data. Labs across the university will be randomly assigned to receive the training, and outcomes of their research will be assessed across multiple years.

Preregistration of research designs and analysis plans is a proposed method to increase the credibility of reported research and a means to increase transparency of the research workflow. However, preregistration is rarely practiced

outside of clinical trials where it is required by law in the U.S. and as a condition for publication in most journals that publish them. Research suggests that preregistration may counter some questionable practices, such as flexible definition of analytic models and outcome variables in order to find positive results (*Kaplan and Irvin, 2015*). Public registration also makes it possible to compare publications and registrations of the same study to identify cases in which outcomes were changed or unreported, as is the focus of the COMPare project based at the University of Oxford (*COMPare, 2016*). Similar efforts include the AllTrials project, run by an international team (*AllTrials, 2016*), and extending beyond just preregistration of planned studies to retroactive registration and transparent reporting for previously conducted clinical trials. Another example is the AsPredicted project, which is run by researchers at the University of Pennsylvania and University of California Berkeley, and offers preregistration services for any discipline (*AsPredicted, 2016*).

To initiate similar research efforts in the basic and preclinical sciences, the Center for Open Science launched the Preregistration Challenge, offering one thousand $1,000 awards to researchers that publish the outcomes of preregistered research (*Center for Open Science, 2016*).

### Openness as a continuum of practices

While there are clear definitions and best practices for open access (*Chan et al., 2002*), open data (*Open Knowledge, 2005*; *Murray-Rust et al., 2010*), and open source (*Open Source Initiative, 2007*), openness is not 'all-or-nothing'. Not all researchers are comfortable with the same level of sharing, and there are a variety of ways to be open (see *Box 1*). Openness can be thus defined by a continuum of practices, starting perhaps at the most basic level with openly self-archiving postprints and reaching perhaps the highest level with openly sharing grant proposals, research protocols, and data in real time. Fully open research is a long-

---

## Box 1. What can I do right now?

Engaging in open science need not require a long-term commitment or intensive effort. There are a number of practices and resolutions that researchers can adopt with very little effort that can help advance the overall open science cause while simultaneously benefiting the individual researcher.

1. **Post free copies of previously published articles in a public repository**. Over 70% of publishers allow researchers to post an author version of their manuscript online, typically 6-12 months after publication (see section "Publish where you want and archive openly").

2. **Deposit preprints of all manuscripts in publicly accessible repositories** as soon as possible – ideally prior to, and no later than, the initial journal submission (see section "Postprints").

3. **Publish in open access venues** whenever possible. As discussed in Prestige and journal impact factor, this need not mean forgoing traditional subscription-based journals, as many traditional journals offer the option to pay an additional charge to make one's article openly accessible.

4. **Publicly share data and materials via a trusted repository**. Whenever it is feasible, the data, materials, and analysis code used to generate the findings reported in one's manuscripts should be shared. Many journals already require authors to share data upon request as a condition of publication; pro-actively sharing data can be significantly more efficient, and offers a variety of other benefits (see section "Resource management and sharing").

5. **Preregister studies**. Publicly preregistering one's experimental design and analysis plan in advance of data collection is an effective means of minimizing bias and enhancing credibility (see section "Open questions"). Since the preregistration document(s) can be written in a form similar to a Methods section, the additional effort required for preregistration is often minimal.

term goal to strive towards, not a switch we should expect to flip overnight.

Many of the discussions about openness center around the associated fears, and we need encouragement to explore the associated benefits as well. As researchers share their work and experience the benefits, they will likely become increasingly comfortable with sharing and willing to experiment with new open practices. Acknowledging and supporting incremental steps is a way to respect researchers' present experience and comfort, and produce a gradual culture change from closed to open research. Training of researchers early in their careers is fundamental. Graduate programs can integrate open science and modern scientific computing practices into their existing curriculum. Methods courses could incorporate training on publishing practices such as proper citation, author rights, and open access publishing options. Institutions and funders could provide skills training on self-archiving articles, data, and software to meet mandate requirements. Importantly, we recommend integrating education and training with regular curricular and workshop activities so as not to increase the time burden on already-busy students and researchers.

## Summary

The evidence that openly sharing articles, code, and data is beneficial for researchers is strong and building. Each year, more studies are published showing the open citation advantage; more funders announce policies encouraging, mandating, or specifically financing open research; and more employers are recognizing open practices in academic evaluations. In addition, a growing number of tools are making the process of sharing research outputs easier, faster, and more cost-effective. In his 2012 book *Open Access*, Peter Suber summed it up best: "*[OA] increases a work's visibility, retrievability, audience, usage, and citations, which all convert to career building. For publishing scholars, it would be a bargain even if it were costly, difficult, and time-consuming. But...it's not costly, not difficult, and not time-consuming.*" (*Suber, 2012*)

### Acknowledgements
This paper arose from the ''Open Source, Open Science'' meeting held March 19-20th, 2015 at the Center for Open Science in collaboration with Mozilla Science Lab. This meeting was supported by the National Institute of Aging (R24AG048124), the Laura and John Arnold Foundation, and the John Templeton Foundation (46545). The authors thank all those who responded to our public calls for comment – especially Virginia Barbour, Peter Binfield, Nazeefa Fatima, Daniel S. Katz, Sven Kochmann, Ehud Lamm, Alexei Lutay, Ben Marwick, Daniel Mietchen, Ian Mulvany, Cameron Neylon, Charles Oppenheim, Pandelis Perakakis, Richard Smith-Unna, Peter Suber, and Anne-Katharina Weilenmann – whose feedback helped us improve this manuscript.

**Erin C McKiernan** is in the Department of Physics, Faculty of Science, National Autonomous University of Mexico, Mexico City, Mexico

http://orcid.org/0000-0002-9430-5221

**Philip E Bourne** is in the Office of the Director, National Institutes of Health, Bethesda, United States

http://orcid.org/0000-0002-7618-7292

**C Titus Brown** is in the Population Health and Reproduction, University of California, Davis, Davis, United States

**Stuart Buck** is in the Laura and John Arnold Foundation, Houston, United States

**Amye Kenall** is in the BioMed Central, London, United Kingdom

http://orcid.org/0000-0002-3030-8001

**Jennifer Lin** is in the CrossRef, Oxford, United Kingdom

**Damon McDougall** is in the Institute for Computational Engineering and Sciences, University of Texas at Austin, Austin, United States

**Brian A Nosek** is in the Center for Open Science, Charlottesville, United States

**Karthik Ram** is in the Berkeley Institute for Data Science, University of California, Berkeley, Berkeley, United States

**Courtney K Soderberg** is in the Center for Open Science, Charlottesville, United States

**Jeffrey R Spies** is in the Center for Open Science, Charlottesville, United States and at the Department of Engineering and Society, University of Virginia, Charlottesville, United States

**Kaitlin Thaney** is in the Mozilla Science Lab, Mozilla Foundation, New York, United States

**Andrew Updegrove** is in the Gesmer Updegrove LLP, Boston, United States

**Kara H Woo** is in the Center for Environmental Research, Education, and Outreach, Washington State University, Pullman, United States and at the Information School, University of Washington, Seattle, United States

http://orcid.org/0000-0002-5125-4188

**Tal Yarkoni** is in the Department of Psychology, University of Texas at Austin, Austin, United States

*Author contributions:* ECM, PEB, CTB, SB, AK, JL, DM, BAN, KR, CKS, JRS, KT, AU, KHW, TY, Conception and design, Drafting or revising the article

*Competing interests:* ECM: Founder of the 'Why Open Research?' project, an open research advocacy and educational site funded by the Shuttleworth Foundation. She is also a figshare and PeerJ Preprints advisor, Center for Open Science ambassador, and OpenCon organizing committee member - all volunteer positions. AK: Works at the open access publisher BioMed Central, a part of the larger SpringerNature company, where she leads initiatives around open data and research and oversees a portfolio of journals in the health sciences. JL: Works for CrossRef and is involved in building infrastructure that supports open science research: Principles for Open Scholarly Research, open data initiatives, and open scholarly metadata. BAN, CKS, JRS: Employed by the non-profit Center for Open Science, which runs the Open Science Framework, and includes in its mission "increasing openness, integrity, and reproducibility of scientific research". KT: Employed by the Mozilla Foundation, where she leads the organization's open science program - the Mozilla Science Lab. The Science Lab supports fellowships, training and prototyping, including work on open research badges. The other authors declare that no competing interests exist.

## Funding

| Funder | Grant reference number | Author |
|---|---|---|
| National Institute on Aging | R24AG048124 | Brian A Nosek Courtney K Soderberg |
| Laura and John Arnold Foundation | | Brian A Nosek Jeffrey R Spies |
| John Templeton Foundation | 46545 | Brian A Nosek Jeffrey R Spies |

The funders had no role in study design, data collection and interpretation, or the decision to submit the work for publication.

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
