## [Decision Letter]

Thank you for submitting your article "The benefits of open research: How sharing can help researchers succeed" to *eLife* for consideration as a Feature Article. Your article has been reviewed by three peer reviewers and the *eLife* Features Editor (Peter Rodgers), and this decision letter has been compiled to help you prepare a revised submission.

All three reviewers have agreed to reveal their identity: Robert Kiley; Chris Gorgolewski; Vincent Lariviere.

General assessment:

This paper is a much-needed overview of benefits and practical advice about open research. Instead of preaching and using moral arguments, the authors focus on benefits to the researcher of doing research openly, and provide evidence that researchers who practice open research (making articles OA, sharing data, publishing code under an open licence etc.) enjoy significant benefits (more citations, more media coverage) compared with researchers who don't practice open research.

Essential revisions:

1) (Section 2.1) Although the authors do provide evidence that OA leads to more citations, I think they need to recognise that there is significant disagreement within the community about this. The paper does cite one study from Davis (Davis, 2011) but he has published quite a few papers on this topic: please consider citing another one of these studies.

2) (Figure 1) It is misleading to plot only the maximum citation advantage: please plot the median advantage and/or the range of advantages (or, alternatively, drop the figure).

3) (Section 2.3) I found this a little self-contradictory. At the start the authors argue (rightly) that IF are a flawed measure, but then go on to quote the IF of OA journals. I know they go to so say "we reiterate that IF are flawed[…]" but if they really believe this, then arguing that some OA journals have high IF doesn't make sense. Please consider deleting the passage "In the 2012 Journal Citation Report...choose between IF and OA)."

4) (Section 2.4; first paragraph) This is debatable and should be toned down a bit. Results obtained with F1000 tend to show the opposite (https://scholarlykitchen.sspnet.org/2013/03/27/how-rigorous-is-the-post-publication-review-process-at-f1000-research/).

5) (Section 2.5.1) I think this section should mention the recent ASAPbio meeting – and subsequent researcher survey – which seem to suggest that researchers in the life sciences have woken up to the potential of preprints (albeit 25 years after physicists reached the same conclusion!)

6) (Section 3.2) Right at the start of the article (Introduction, first paragraph) the authors get to the nub of the problem – namely that open practices could present a risk to career advancement. This in my opinion is the big issue. Until researchers are persuaded that making their outputs open is *not* going to adversely impact them (and ultimately come to believe that it will benefit them) then changing behaviour is always going to be difficult. As such I was surprised that section 3.2 seemed to suggest that Funder mandates are sufficient to bring about this change. Although I agree that some mandates are important, on their own they are not sufficient.

The article would be improved if it recognised that mandates are not enough and then set out a list of things funders could do to help move the needle in this space (e.g., maybe end-of-grant reports should actively recognise and reward behaviours like data sharing, undertaking peer review, publishing papers on preprint servers etc. – and be less fixated on counting journal article outputs.)

Further to this: the NIH requires grant applications to include a data-management plans: however, when the NIH is considering grant applications, it does not take into account if applicants have a history of sharing data, and it does not penalize applicants if data from previous grants have not been shared. Until data sharing becomes an important part of review procedure, change will remain slow.

7) (Section 2.7) To be transparent, please mention the higher APCs of PLOS journals, as well as of for-profit publishers like Elsevier, Wiley, Springer, etc. When taking about high APCs the authors may also wish to cite the data the Wellcome Trust (and others) have published on this topic (e.g. blog.wellcome.ac.uk/2016/03/23/wellcome-trust-and-coaf-open-access-spend-2014-15/).

8) (Section 4.2) Being devil’s advocate here: the authors report that research projects that share data produce twice as many publications as those that do not share data. Isn't the more likely explanation of this relation that successful projects that have published a lot can afford to share data because the risk and consequences of scooping are much lower/smaller?

9) (Table 1) Please add the following columns: "Ability to leave feedback", "Provides DOI", and "Indexed by Google Scholar".

10) (Figure 2) Please expand the caption for this figure to better explain the five different organisations shown in the figure, maybe by giving examples of each type of organization. Please also explain why there are categories for "research organisations", "funders" and "funders and research organisations". Also, please explain the category "multiple research organisations"

---

## [Author Response]

*Essential revisions:*

1) (Section 2.1) Although the authors do provide evidence that OA leads to more citations, I think they need to recognise that there is significant disagreement within the community about this. The paper does cite one study from Davis (Davis, 2011) but he has published quite a few papers on this topic: please consider citing another one of these studies.

To present this information in a more balanced way, we deleted the word ‘overwhelming’ from Section 2.1, first paragraph, and made other minor changes to wording of this section.

We added another paper from Davis (Davis, 2008):

P.M. Davis, B.V. Lewenstein, D.H. Simon, J.G. Booth, and M.J.L. Connolly. Open access publishing, article downloads, and citations: randomised controlled trial. BMJ, 337:a568, 2008.

We also added citations to 3 other studies which failed to find an OA citation advantage:

T.F. Frandsen. The effects of open access on un-published documents: A case study of economics working papers. Journal of Informetrics, 3(2):124–133, 2009.

P. Gaule and N. Maystre. Getting cited: does open access help? Research Policy, 40(10): 1332–1338, 2011.

V.C. Lansingh and M.J. Carter. Does open access in ophthalmology affect how articles are subsequently cited in research? Ophthalmology, 116(8):1425–1431, 2009.

We expanded Figure 1 to include some studies in which no OA citation advantage, or even a disadvantage, was found for certain disciplines (see below).

2) (Figure 1) It is misleading to plot only the maximum citation advantage: please plot the median advantage and/or the range of advantages (or, alternatively, drop the figure).

We revised Figure 1 to include mean citation advantages (medians are often not reported). These are now shown as a relative citation rate, instead of percentage. We also expanded the figure to include more disciplines and more studies, including some in which an OA citation advantage was not found.

3) (Section 2.3) I found this a little self-contradictory. At the start the authors argue (rightly) that IF are a flawed measure, but then go on to quote the IF of OA journals. I know they go to so say "we reiterate that IF are flawed[…]." but if they really believe this, then arguing that some OA journals have high IF doesn't make sense. Please consider deleting the passage "In the 2012 Journal Citation Report…choose between IF and OA)."

We appreciate the reviewer’s concern, but believe strongly that we have to discuss IF since it is often cited by researchers as a barrier to publishing openly. In numerous author surveys, researchers repeatedly rank impact factor and associated journal reputation as among the most important factors they consider when deciding where to publish. To support this, we added citations to the following author surveys:

Nature Publishing Group (2015): Author Insights 2015 survey. figshare. https://dx.doi.org/10.6084/m9.figshare.1425362.v7

Solomon DJ. (2014) A survey of authors publishing in four megajournals. PeerJ 2:e365 https://doi.org/10.7717/peerj.365

Given our researcher-centric approach, it is important to recognize concerns about IF as a practical, albeit regrettable, reality. We believe that ignoring this reality, and specifically removing the recommended passage with data on the IFs of OA journals, would weaken the paper. This information provides researchers with options that satisfy both their worry about publishing in high IF journals and their wish to do so openly. To clarify our goals with this section, we made several small changes in wording and sentence order, and added a closing statement, which reads:

“We hope that in the next few years, use of IF as a metric will diminish or cease entirely, but in the meantime, researchers have options to publish openly while still meeting any IF-related evaluation and career-advancement criteria.

4) (Section 2.4; first paragraph) This is debatable and should be toned down a bit. Results obtained with F1000 tend to show the opposite (https://scholarlykitchen.sspnet.org/2013/03/27/how-rigorous-is-the-post-publication-review-process-at-f1000-research/).

We reworded this section to read:

“Some studies suggest open peer review may produce reviews of higher quality, including better substantiated claims and more constructive criticisms, compared to closed review [Donovan, Watson and Osborne, 2015; McCabe and Snyder, 2015]. Additional studies have also argued that transparent peer review processes are linked to measures of quality. Other studies have reported no differences in the quality of open versus closed reviews [Wicherts, 2016; Rooyen et al., 1999]. More research in this area is needed.”

We have added the following references to controlled studies finding no difference in quality of open versus closed reviews:

S. Van Rooyen, F. Godlee, S. Evans, N. Black, and R. Smith. Effect of open peer review on quality of reviews and on reviewers’ recommendations: a randomised trial. BMJ, 318(7175):23–27, 1999.

S. van Rooyen, T. Delamothe, and S.J.W. Evans. Effect on peer review of telling reviewers that their signed reviews might be posted on the web: randomised controlled trial. BMJ, 341:c5729, 2010.

5) (Section 2.5.1) I think this section should mention the recent ASAPbio meeting – and subsequent researcher survey – which seem to suggest that researchers in the life sciences have woken up to the potential of preprints (albeit 25 years after physicists reached the same conclusion!)

We added mention of the ASAPbio meeting (end of Section 2.5.1) and cited the following references on the meeting and survey:

J.M. Berg, N. Bhalla, P.E. Bourne, M. Chalfie, D.G. Drubin, J.S. Fraser, C.W. Greider, M. Hen- dricks, C. Jones, R. Kiley, S. King, M.W. Kirschner, H.M. Krumholz, R. Lehman, M. Leptin, B. Pulverer, B. Rosenzweig, J.E. Spiro, M. Stebbins, C. Strasser, S. Swaminathan, P. Turner, R.D. Vale, K. VijayRaghavan, and C. Wolberger. Preprints for the life sciences. Science, 352(6288): 899–901, 2016.

ASAPbio. Opinions on preprints in biology. Accessed May, 2016 at http://asapbio.org/survey. Data available via figshare https://dx.doi.org/10.6084/m9.figshare.2247616.v1.

6) (Section 3.2) Right at the start of the article (Introduction, first paragraph) the authors get to the nub of the problem – namely that open practices could present a risk to career advancement. This in my opinion is the big issue. Until researchers are persuaded that making their outputs open is not going to adversely impact them (and ultimately come to believe that it will benefit them) then changing behaviour is always going to be difficult. As such I was surprised that section 3.2 seemed to suggest that Funder mandates are sufficient to bring about this change. Although I agree that some mandates are important, on their own they are not sufficient.

While we agree that mandates are unlikely to bring about the culture change we would like to see, there is evidence that mandates are effective in increasing rates of article and data sharing (see work from Harnad and colleagues, especially). More importantly, our goal with this section is not to argue that mandates are sufficient, but rather that “[researchers] already engaging in [open] practices will likely have a competitive advantage”.

The article would be improved if it recognised that mandates are not enough and then set out a list of things funders could do to help move the needle in this space (e.g., maybe end-of-grant reports should actively recognise and reward behaviours like data sharing, undertaking peer review, publishing papers on preprint servers etc. – and be less fixated on counting journal article outputs.)

We recognize in the subsequent section (section 4) that “[funder mandates] may be a compelling but dissatisfying reason to practice openly”. However, our primary target audience for this article is researchers, so we have focused on outlining the steps they can take and showing them “there are other compelling reasons to share”.

Further to this: the NIH requires grant applications to include a data-management plans: however, when the NIH is considering grant applications, it does not take into account if applicants have a history of sharing data, and it does not penalize applicants if data from previous grants have not been shared. Until data sharing becomes an important part of review procedure, change will remain slow.

We added mention of policy revisions implemented by NIH and Wellcome Trust, detailing how funds can be suspended or withheld if researchers do not comply with mandates (Section 3.2, last paragraph). We cited the following relevant references, one of which (van Noorden, 2014) discusses how both funders have already followed through on enforcement:

National Institutes of Health (NIH). Upcoming Changes to Public Access Policy Reporting Requirements and Related NIH Efforts to Enhance Compliance, 2012. Retrieved June, 2016 from http://grants.nih.gov/grants/guide/notice-files/NOT-OD-12-160.html. Last updated Feb., 2013.

Van Noorden, R. Funders punish open-access dodgers. Nature News, 2014. Retrieved June, 2016 from http://www.nature.com/news/funders-punish-open-access-dodgers^-1^.15007.

Wellcome Trust. Wellcome Trust strengthens its open access policy, 2012. Retrieved June, 2016 from https://wellcome.ac.uk/press-release/wellcome-trust-strengthens-its-open-access-policy.

7) (Section 2.7) To be transparent, please mention the higher APCs of PLOS journals, as well as of for-profit publishers like Elsevier, Wiley, Springer, etc. When taking about high APCs the authors may also wish to cite the data the Wellcome Trust (and others) have published on this topic (e.g. blog.wellcome.ac.uk/2016/03/23/wellcome-trust-and-coaf-open-access-spend-2014-15/).

We added mention of the higher APCs charged by some OA publishers, like PLOS and Frontiers (Section 2.7, last paragraph). We also felt the no-cost/low-cost examples here were numerous, so we have stricken two of them.

We added the suggested reference from Wellcome Trust, as well as one from Stuart Lawson documenting high APCs:

S. Lawson. APC data for 27 UK higher education institutions in 2015. figshare, 2016. Retrieved June, 2016 from https://dx.doi.org/10.6084/m9.figshare.1507481.v4.

Wellcome Trust. Wellcome Trust and COAF Open Access Spend, 2014-15,. Retrieved June, 2016 from https://blog.wellcome.ac.uk/2016/03/23/wellcome-trust-and-coaf-open-access-spend-2014-15/. Data available via figshare doi:10.6084/m9.figshare.3118936.v1.

We also added a citation to a new study of over 10, 300 OA journals, showing 71% do not charge an APC (Section 2.7, first paragraph), and that the average APC for OA journals is around $665 (Section 2.7, second paragraph):

W. Crawford. Gold Open Access Journals 2011-2015. Cites & Insights Books, 2016. Accessed June, 2016 via http://waltcrawford.name/goaj.html.

8) (Section 4.2) Being devil’s advocate here: the authors report that research projects that share data produce twice as many publications as those that do not share data. Isn't the more likely explanation of this relation that successful projects that have published a lot can afford to share data because the risk and consequences of scooping are much lower/smaller?

The authors of Pienta et al. admit that “It is unclear whether larger numbers of primary publications lead to data sharing or if sharing data leads to more primary publications”. However, the authors did control for factors such as Principal Investigator age, gender, career status, and funding history, as well as features of the grant such as duration as an indirect measure of grant size. None of these factors sufficiently explained the primary or secondary publication advantage conferred by data sharing.

9) (Table 1) Please add the following columns: "Ability to leave feedback", "Provides DOI", and "Indexed by Google Scholar".

We added the columns “Can leave feedback?” and “Third party persistent ID?”. The latter is broader and includes externally managed persistent identifiers such as DOIs, Handles, and ARKs. We added a footnote to the table saying that all the listed preprint servers and repositories are indexed by Google Scholar. After community feedback, we also added several relevant repositories.

10) (Figure 2) Please expand the caption for this figure to better explain the five different organisations shown in the figure, maybe by giving examples of each type of organization. Please also explain why there are categories for "research organisations", "funders" and "funders and research organisations". Also, please explain the category "multiple research organisations"

Based on the information provided by ROARMAP, we added the following explanation and examples to the figure caption:

“Data are broken down by policymaker type: funder (e.g. Wellcome Trust), joint funder and research organization (e.g. British Heart Foundation), multiple research organizations i.e. associations and consortia (e.g. Max Planck Society), research organization i.e. university or research institution (e.g. CERN), and subunit of research organization (e.g. Columbia University Libraries).”